# Characterization and Roles of Membrane Lipids in Fatty Liver Disease

**DOI:** 10.3390/membranes12040410

**Published:** 2022-04-09

**Authors:** Morgan Welch, Cassandra Secunda, Nabin Ghimire, Isabel Martinez, Amber Mathus, Urja Patel, Sarayu Bhogoju, Mashael Al-Mutairi, Kisuk Min, Ahmed Lawan

**Affiliations:** 1Department of Biological Sciences, University of Alabama in Huntsville, Huntsville, AL 35899, USA; mmw0025@uah.edu (M.W.); cs0244@uah.edu (C.S.); ng0063@uah.edu (N.G.); inm0003@uah.edu (I.M.); akm0027@uah.edu (A.M.); up0006@uah.edu (U.P.); 2Department of Internal Medicine, University of Kentucky, Lexington, KY 40536, USA; sb0213@uah.edu; 3Department of Medical Laboratory Sciences, Faculty of Allied Health Sciences, Kuwait University, Sulaibekhat 90805, Kuwait; mashael.almuteiri@ku.edu.kw; 4Department of Kinesiology, University of Texas at El Paso, El Paso, TX 79968, USA; kmin@utep.edu

**Keywords:** obesity, membrane lipids, fatty liver disease, insulin resistance, metabolic syndrome

## Abstract

Obesity has reached global epidemic proportions and it affects the development of insulin resistance, type 2 diabetes, fatty liver disease and other metabolic diseases. Membrane lipids are important structural and signaling components of the cell membrane. Recent studies highlight their importance in lipid homeostasis and are implicated in the pathogenesis of fatty liver disease. Here, we discuss the numerous membrane lipid species and their metabolites including, phospholipids, sphingolipids and cholesterol, and how dysregulation of their composition and physiology contribute to the development of fatty liver disease. The development of new genetic and pharmacological mouse models has shed light on the role of lipid species on various mechanisms/pathways; these lipids impact many aspects of the pathophysiology of fatty liver disease and could potentially be targeted for the treatment of fatty liver disease.

## 1. Introduction

Obesity predisposes to the development of cardiovascular disease, type 2 diabetes, nonalcoholic fatty liver disease (NAFLD) and atherosclerosis [1,2,3]. Nonalcoholic fatty liver disease (NAFLD) is a chronic liver disease that is characterized by excessive lipid accumulation in hepatocytes in the absence of substantial alcohol ingestion, viral hepatitis or other liver disorders [4,5]. Nearly one third of adult Americans suffer from NAFLD [5,6], and over 600 million people have NAFLD worldwide [5,7]. It is estimated that the more severe form of NAFLD, non-alcoholic steatohepatitis (NASH) is expected to become the number one reason for liver transplantation in the US [7,8]. It has been projected that the annual medical cost for NAFLD/NASH patients in the United States is over USD 100 billion and approximately EUR 35 billion for United Kingdom, France, Germany and Italy [8,9]. Hyperlipidemia and fatty liver disease are two major diseases for which there is no effective therapy available. Although the most demonstrated treatment strategies for NAFLD are weight loss and exercise, treatment parameters for these approaches often are not attainable, so a number of therapeutic agents are under active investigation. Statins have improved hepatic steatosis in patients with NAFLD/NASH. However, statins have not been shown to ameliorate hepatic inflammation and fibrosis in human NASH. The majority of these agents target either lipid accumulation pathways in the hepatocyte or else the inflammation and hepatocyte injury that results from steatosis-induced metabolic stress, and most basic research in NAFLD focuses on mechanisms responsible for these pathophysiological endpoints.

The primary defect in the pathogenesis of NAFLD is dysregulated lipid metabolism and the excessive accumulation of fatty acids and cholesterol within hepatocytes known as hepatic lipotoxicity [6]. Lipids play very important role in cellular functions and are major components of biological membranes. Cell membranes are the barriers that delineate the various organelles of the cells into separate but interrelated sections. It is a lipid bilayer and contains proteins that act as integral, peripheral and transmembrane proteins. The major lipids found in the cell membrane of eukaryotes are phospholipids, sphingolipids, cholesterol and sterol lipids [10]. In mammals, phospholipids are formed from two hydrophobic fatty acyl chains and one hydrophilic group, sphingolipids are synthesized from ceramides as the main precursors, while cholesterol with a steroid backbone is the main sterol component of the membrane [11]. Carbohydrates interact with both lipids and proteins to form glycolipids and glycoproteins, respectively [12]. Apart from serving as the major components of cell membrane and precursors for many hormones, membrane lipids play very important functions in lipid homeostasis, cellular signaling, energy storage and provide an appropriate environment for protein function [10,13]. The organizational and functional properties of cell membranes is unique considering that the composition of lipids differs between the cell membrane and membrane surrounding cell organelles or other compartments [14]. More studies are required to investigate how diverse membrane lipids interact with proteins and regulate their functions.

The liver plays a major role in the metabolism of lipids. Although hepatocytes are the predominant cells in the liver, other cells including Kupffer cells, hepatic stellate cells and cholangiocytes contribute to hepatic function in health and disease. Subsequently, all these cells are subject to injury resulting from hepatic steatosis and lipotoxicity driving the development of NAFLD/NASH. Many studies have shown that NAFLD is linked with changes in intracellular lipid composition in the liver [6]. Furthermore, most of these studies including genetic and pharmacological focused on the role of accumulation of triglycerides as the main driver for NAFLD [5]. Considering that the pathophysiology of NAFLD is complex, accumulation of triglyceride alone is not sufficient to cause NAFLD. The basis of the review is to emphasize this emerging area of lipid research that has examined the role different lipid species including, phospholipids, sphingolipids and cholesterol in the development and progression of NAFLD/NASH.

## 2. Role of Sphingomyelin-Mediated Ceramide in Fatty Liver Disease

Sphingolipids are important constituents of cell membrane acting both intracellularly and extracellularly to regulate cell proliferation, differentiation, cell death and immunological responses [15]. Sphingomyelin is also an important structural component of biological membrane and is one of the endpoints in the synthesis of sphingolipids, which are a class of membrane lipids for vital structural and signaling bioactive molecules [16,17]. Ceramides are members of the sphingolipid family and are one of the major lipid constituents in the lipid bilayer of the cell membrane [18]. Since ceramides and sphingomyelin are important components in the double membrane-bound sphingolipids, dysregulated metabolisms of ceramide and sphingomyelin play a critical role in pathologies of the liver, including NAFLD and NASH [19]. Ceramides are generated through three pathways that include de novo synthesis, sphingomyelin hydrolysis, and salvage pathway. The de novo synthesis of ceramide begins with the condensation of serine and palmitoyl-CoA through serine palmitoyltransferase, followed by the activity of 3-ketodihydrosphingosin reductase, dihydroceramide synthase, and dihydroceramide desaturase [20]. The de novo synthesis of ceramide occurs in the endoplasmic reticulum. Then, ceramide is subsequently transported to the Golgi apparatus to be metabolized to other sphingolipids, such as sphingomyelin [18]. Ceramide is also generated by the sphingomyelin hydrolysis through sphingomyelinases [20,21]. In the salvage pathway, ceramide generated by sphingomyelin hydrolysis is further hydrolyzed by ceramidases to sphingosine, which is reacylated through ceramide synthases to regenerate ceramide [22].

### 2.1. Hepatotoxicity of Ceramides in the Liver

Ceramides exert biological effects through cellular proliferation, differentiation, and cell death [23]. Ceramides regulate several pathways involved in insulin resistance, oxidative stress, inflammation, and apoptosis, all of which are associated with pathogenesis of fatty liver disease [23,24]. Growing evidence suggest that the inhibition of the de novo ceramide synthesis reduced hepatic lipid accumulation [25]. Recent studies also demonstrated that reduction in ceramide synthesis can improve steatosis and insulin resistance [26,27]. In these studies, the reduction in ceramide synthesis using both pharmacologic and genetic models of ceramide synthase reduction prevents lipid droplet accumulation and insulin resistance in the experimental models of NAFLD (Figure 1). The prevention of steatosis and improvement of insulin resistance with the pharmacologic inhibition of ceramide synthesis are associated with reduced perilipin 2 (PLIN2), which is a lipid-droplet protein and is up-regulated in alcoholic steatosis [26]. Another study also demonstrated that treatment of cultured hepatocytes with proinflammatory cytokines such as interleukin-1β (IL-1β) causes a rapid turnover of both sphingomyelin as well as intracellular ceramide content [18,28]. Ichi I et al. revealed that induction of necrosis by carbon tetrachloride results in significant increases in ceramide concentration in both plasma and the liver [29]. The level of sphingomyelin with distinct saturated acyl chains (C18:0, C20:0, C22:0, and C24:0) is upregulated in obese individuals and is highly associated with the development of insulin resistance, and dyslipidemia [30]. Furthermore, obese individuals diagnosed with NAFLD showed high level of sphingomyelin with the saturated acyl chains [30]. These studies demonstrate obesity promote the accumulation of ceramide in the liver thereby inducing insulin resistance, hepatic steatosis and hepatocyte apoptosis (Figure 1).

In human studies, it has been shown that ceramide and sphingomyelin concentrations in the liver of obese people are greater than in the subcutaneous and intra-abdominal adipose tissues [31]. Another study revealed that lifestyle-induced weight reduction lowers hepatic ceramide expression in obese patients with NAFLD and that lower serum ceramides are associated with decreased ceramide gene expression in the liver [32].

Along with ceramides, its metabolites such as gangliosides and glucosylceramides are also involved in the development of insulin resistance. Although no previous research has found a direct link between those metabolites and NAFLD, their roles in insulin signaling and resistance suggest that they may play a role in the progression of NAFLD. GM3 is a ganglioside that interacts directly with IRS-1, the first and limiting step in the insulin signaling cascade, to increase insulin resistance. Therefore, it reduces autophosphorylation of the insulin receptor by inhibiting tyrosine phosphorylation of IRS-1 [33,34]. Additionally, GM3 was proposed to work as an intermediate between the inflammatory state and insulin resistance. GM3 accumulation was detected in lipid rafts by TNF-α stimulation [34,35]. Ceramide-1-phosphate, another metabolite that is produced by phosphorylation of ceramide through ceramide kinase, also contributes to insulin resistance. Ceramide kinase deletion increased insulin sensitivity in mice fed a high-fat diet and reduced macrophage infiltration in adipocytes [36]. Taken together, these studies provide evidence that ceramide and its metabolites represent a risk for the development of NAFLD and further work is needed to understand the regulation of ceramides and signaling in the progression of NAFLD.

### 2.2. Ceramides and Hepatic Inflammation

Fatty liver disease is characterized by high level of inflammation. Recently, a crucial role of ceramide in hepatic inflammation has been reported. Ceramides have been shown to promote inflammation through interaction with pro-inflammatory cytokines. Pro-inflammatory cytokines such as TNF-α induce ceramide synthesis by activating sphingomyelinases in hepatocytes [15,17,37]. Park J.W. et al. demonstrated that overexpression of ceramide synthases elevates the secretion of inflammatory cytokines, including TNF-α, IL-1β, and IL-6 in Hep3B cells [28]. Interestingly, the suppression of ceramide generation attenuated high-fat-diet-induced hepatic cytokine production [28]. Other studies also established that the inhibition of sphingomyelinase attenuates hepatic inflammation [19,38]. The interaction between ceramides and inflammatory cytokines promotes the recruitment of inflammatory cells to the liver, leading to hepatic inflammation [20]. Furthermore, ceramide-mediated inflammation increases mitochondrial oxidative stress, promoting apoptosis in hepatocyte [39]. Interestingly, the treatment with adiponectin, which acts as an anti-inflammatory cytokine secreted by adipose tissue, resulted in a decrease in ceramide levels in the liver that was accompanied with increased ceramidase activity [22]. Taken together, there is evidence that ceramide may play a central role in hepatic inflammation through the interaction with cytokines (Figure 1).

### 2.3. Ceramides and Hepatocyte Cell Death

The sphingomyelin-mediated ceramide pathway has emerged as a key regulator in cellular apoptosis [40]. Ceramide has been shown to increase mitochondrial permeability transition, leading to caspase-3 activation and cell death. Further, ceramide-induced hepatic apoptosis is enhanced with hydrogen peroxidase [41]. Jin J. et al. showed that upregulation of ceramides through the activation of acid sphingomyelinase is required for Bax translocation from the cytoplasm to mitochondria, leading to cytochrome c release and apoptosis [42]. Evidence also shows that the activation of the de novo synthesis of ceramide promotes hepatocyte cell death through inflammatory cytokines and membrane death receptors [26,27]. Conversely, inhibition of ceramide synthesis attenuates apoptosis induced by pro-inflammatory cytokines. Indeed, Sphingomyelinase-deficient mice exhibited resistance to TNF-α-mediated hepatocellular apoptosis and liver damage [43]. Evidence also demonstrates that hepatic inflammation and apoptosis is significantly attenuated in HFD-fed rats treated with myriocin, which inhibits ceramide synthesis and lipid accumulation. The rats also exhibit improved hepatic steatosis and fibrosis [30]. Collectively, these studies demonstrate that sphingomyelin-mediated ceramide pathways regulate hepatocyte cell death through mitochondrial apoptosis and/or inflammatory cytokine activity (Figure 1).

### 2.4. Ceramides and Hepatic Fibrosis

Hepatic fibrosis is one of the major consequences of fatty liver disease [44]. Hepatic fibrosis results from the excessive accumulation of extracellular matrix protein such as collagen fibers, fibronectins, and proteoglycans that lead to cirrhosis and liver failure [45]. Several studies have reported that ceramide biosynthesis contributes to the development of hepatic fibrosis [30,46]. High fat diet-fed rats exhibited increased expression of fibrosis markers such as alpha smooth muscle actin (α-SMA) and collagen type I alpha 2 (COL1A2) in the liver, whereas rats treated with myriocin attenuated the expression of the fibrosis markers during HFD [30]. Evidence shows that hepatic stellate cells with activation of fibroblasts exhibited accumulation of specific ceramide species and ceramide synthases [47]. Moles A. et al. demonstrated that acid sphingomyelinase is activated during the differentiation of hepatic stellate cells into fibroblasts with ceramide accumulation [48]. However, the role of ceramide and sphingomyelin in liver fibrosis remains controversial with inconsistent results. A previous study showed that ceramide decreases the expression of collagen α-1 in hepatic stellate cells [49]. Li Z. et al. also demonstrated that liver sphingomyelin synthase 1 deficiency causes the activation of TGF-β1, leading to collagen 1αa production in hepatic stellate cells [50]. Collectively, sphingomyelin-mediated ceramide pathway contributes to the development of fibrogenesis in the liver. Given inconsistent results showing the role of ceramide in the liver fibrosis, further studies are needed to identity the molecular mechanisms of how ceramide is involved in the development of hepatic fibrosis.

## 3. Role of Glycolipids in NAFLD

Glycolipids are a type of complex carbohydrate that contains both a glycan and a lipid component [51]. They are important components of cellular membranes and may act as receptors, are involved in cell aggregation and dissociation, and are responsible for cellular interaction and signal transmission [51]. The glyco-components are mono- or oligosaccharide chains that are commonly branched linked to ceramide or other glycerol derivatives and may be replaced with acetyl or sulfate groups. They are classified as glycosphingolipids (GSLs) or glycoglycerolipids, based on the structure of the lipid moiety. GSLs comprise about 100,000 species divided into four subfamilies (phosphosphingolipids, glycosphingolipids, sphingoid bases, and ceramides), all of which have the sphingoid structure [23,52,53]. They are a broad range of biomolecules consisting of both hydrophilic as well as hydrophobic domains that are embedded in the cell membranes of all eukaryotic organisms [13], and are believed to be involved in regulating major cellular processes [54]. As it is well known that NAFLD is linked to a variety of lipid abnormalities in the liver [55], sphingolipid has received a lot of interest in this respect. In recent years, there has been growing interest in the role of glycolipids, particularly sphingolipids, during NAFLD progression [56]. They are thought to have a role in the development of NAFLD through a variety of mechanisms, including obesity, inflammation, insulin resistance, and oxidative stress. The liver has a far greater concentration of sphingolipids, particularly ceramide and sphingomyelin, than any other tissue [31,57]. As a result, the liver is vulnerable to sphingolipotoxicity. Studies have shown that hepatic ceramide and sphingomyelin levels are considerably higher in the livers of rats fed a high-fat diet (HFD) or mice overexpressing acyl-CoA: Diacylglycerol acyltransferase 2 in hepatocytes [58,59,60,61]. At the ceramide genetic level, heterozygous CerS6 knockout mice (CerS6+/) had higher Beta-oxidation and lower CD36/FAT expression, leading to lower lipid accumulation [62]. Furthermore, NAFLD, at the transcription level, is linked to increased expression of genes implicated in three distinct pathways that lead to ceramide formation (de novo synthesis, sphingomyelin hydrolysis, and the salvage route) [58,63]. Recent studies investigated the effects of some natural products on glycolipid disorders amelioration of fatty liver disease. To examine the effect of beta-glycophospholipid in improving liver injury, Zigmond E. et al. utilized an animal model of NASH-*Psammomys obesus* fed on high diet [64]. They reported that beta-glycophospholipid administration in these mice reduced cholesterol and TGs levels and improvement in liver injury [64]. Another study investigated the role of epigallocatechin-3-gallate (EGCG) on glycolipid disorder [65]. Administration of EGCG in a diabetic mouse model demonstrate that this natural product improved the incidence of glycolipid disorder by restoring normal lipid metabolism [65]. Other studies showed that tauroursodeoxycholic acid (TUDCA) ameliorates glycolipid disorder in obese mice by reversing impaired autophagy [66]. These studies demonstrate key role for glycolipid metabolism disorder in the pathogenesis of fatty liver disease and could be potential therapeutics for this disease and other metabolic abnormalities.

## 4. Phosphatidylcholine and Phosphatidylethanolamine-Mediated NAFLD

Phosphatidylcholine (PE) and phosphatidylethanolamine (PC) comprise the vast majority of the phospholipids present in the mammalian cells and as such constitute the major structural components of the cell membrane [67,68]. In the liver, PC is derived from choline through the CDP-choline pathway and conversion from PE to PC through three methylation reactions catalyzed by the PE N-methyltransferase (PEMT) [69,70]. The integrity of the plasma membrane is maintained by the ratio of PC and PE. Therefore, abnormalities in this structural property of the membrane predispose the hepatic cellular membrane to various insults including toxic lipids, immune messenger and activators leading to NAFLD, inflammation and cell death [68] (Figure 2).

### 4.1. Decreasing PE/PC Ratio Results in NAFLD

Nonalcoholic fatty liver disease (NAFLD) presents itself when the hepatic molar ratio of PE/PC either increases or decreases. When gene *Pcyt1a* that encodes the rate limiting enzyme in hepatic PC synthesis (CTP: phosphocholine cytidylyltransferase-α, or CTα) is knocked out in mice, there is a food- and sex-dependent effect on hepatic lipid metabolism [68,71]. Female knockout mice given normal chow had a 20% lower hepatic PC level compared to wild type mice [72]. When knockout mice were challenged with a high fat diet, the sex difference was eliminated, and the effects of the low hepatic PE/PC ratio resulted in the development of NAFLD and NASH [72]. There was also a significant increase in TAG proteins, a result of hepatic PC being eliminated. PC is a critical step in the creation and secretion of TAGs (VLDLs) and the increase in hepatic intercellular TAG is linked to hepatic steatosis and NAFLD [72]. PC is a critical step in the synthesis and secretion of TAGs (VLDLs) and the increase in hepatic intracellular TAG is linked to hepatic steatosis and NAFLD [73]. Similar effects were seen in PEMT−/− mice with a decrease in the PC/PE molar ratio and increase in intracellular TAG as result of depleting phosphatidylethanolamine N-methyltransferase (PEMT), a catalyst for a hepatic specific PC production pathway [74]. Additionally, when PEMT−/− mice were put on a choline deficient diet, intracellular PC levels were reduced by 50% and extracellular PE went up as well to try and maintain homeostasis. The combination of these two events results in the destabilization of the cell membrane and ultimately liver failure (Figure 2). It was also reported that feeding PEMT−/− mice a choline deficient diet prevents them from using the CDP-pathway to convert PE to PC [75,76]. The decrease in the PE:PC ratio in these mice increases membrane permeability and facilitates hepatocyte damage. Knocking out mdr2−/− a gene that codes for a PC-specific flippase (multiple drug resistant protein 2), with PEMT−/− will actually somewhat “rescue” the phenotype, as the PC is no longer being shuttled out of the cell [77]. Of note, hepatic cells with this genotype will catabolize PE and PC in order to maintain appropriate homeostatic levels. Furthermore, MAT1A is a vital enzyme in SAMe (S-adenosylmethionine) synthesis, which is in turn a critical enzyme to the PEMT pathway of PE to PC conversion [78]. Mice with this gene deleted (*MAT1A*−/−) show a decrease in PC production, which in turn affects VLDL production [78]. These decreases are followed by increased amounts of hepatic triglyceride (TG) as well as the activation of adenosine monophosphate-activated protein kinase (AMPK) [78]. The decrease in PE/PC ratio disrupts membrane integrity and opens the cell to various cell injuries and ultimately NAFLD. Leonardi R. et al., reported that silencing of the gene encoding CTP (*PCYT2*) causes a disruption in hepatic intracellular PE production, leading to a 50% decrease in PE levels [79]. Consequently, DAG levels are significantly increased due to the CTP deletion collapsing the CDP-ethanolamine pathway and eliminating DAG to PE conversion [79]. High amounts of DAG levels lead to lipid droplet formation, a common feature of NAFLD.

Mitochondrial protein mitofusin 2 (Mfn2) is a mitochondrial membrane protein that connects ER membranes to mitochondria. In the ER, PS is converted to PE, which is then sent into the mitochondria for modification to PC. Knocking out this membrane (L-KO) protein causes a NAFLD/NASH phenotype due to the halting of PE and PC synthesis [80] (Figure 2).

### 4.2. Increasing PE/PC Ratio Results in NAFLD

Studies indicate that reduction in the PC/PE ratio are not the only way NAFLD pathologies can occur. GMNT (glycine N-methyltransferase) suppresses AdoMet, an important methylator for the PEMT associated PC synthesis pathway. Studies demonstrate that in *GMNT−/−* knockout mice, hepatic PC levels were slightly increased but PE levels were significantly decreased [81,82]. Simultaneously, DAG levels were also higher in the knockout mice, implying that the liver tries to achieve homeostasis between PE and PC by shunting the resultant excess PC into a DAG synthesis pathway [81,82]. This increase in the PC/PE ratio leads to hepatic steatosis. Additionally, GNMT acts as a SAMe suppressor. When this suppressor is removed via knockout (*GNMT−/−*), SAMe amounts naturally increase [81,82]. This significant increase in SAMe increases PC produced in the PEMT pathway. The increased PC levels then cause the liver to produce and secrete more VLDL and HDL in order to maintain homeostatic levels of PC to PE. There is also enhanced PC catabolism and mobilization out of the cell. Two products of PC’s catabolism are DGs (diglycerides) and TGs (triglycerides), which, when at an increased rate in hepatic cells, can lead to endoplasmic reticulum stress and ultimately NAFLD. Taken together, increasing SAMe increases the PE/PC ratio and through a molecular cascade of effects leads to NAFLD.

Collectively, the core of the relationship between NAFLD and PC/PE is in its molar ratio. There must be the proper ratio of the two phospholipids in the membrane, or hepatic cellular membrane integrity will be compromised. This structural issue can let in various lipotoxic molecules, immune messengers and activators, which can lead to cell inflammation/death and NAFLD. Additionally, when there is a disruption in the PE/PC ratio there can be buildups of DAG and/or TAG, which are common causes/symptoms of NAFLD. The main pathways that tend to be altered in NAFLD phenotypes are the hepatic PEMT and choline pathways (Figure 2). Future basic and translational studies are needed to define the mechanistic role of PC and PE in this metabolic disease.

## 5. Oxidized Phospholipids Contribution to NAFLD/NASH

Oxidized PC and PE constitute the major oxidized phospholipids (OxPLs) and they are generally synthesized in platelets, neutrophils and monocytes [83,84,85]. In addition, in macrophages, oxidized cholesteryl esters derived from 15-lipoxygenase can generate OxPLs by transferring the oxidized fatty acyl group to phospholipids [85]. Early studies indicate an essential role of myeloperoxidase-derived oxidative stress in forming oxidized phospholipids in distinct localization in the liver thereby promoting the progression of fatty liver disease [86]. Recently, Sun X. et al. reported that amylin liver NASH (AMLN)-fed Ldlr−/− mice exhibit increased plasma oxidized phospholipids (OxPLs), hepatic steatosis, inflammation and fibrosis compared with mice fed a control diet [87]. They utilized a mouse model that express a functional single-chain variable fragment of a natural antibody that neutralizes OxPLs, called E06, to determine whether neutralization of oxidized phospholipid alleviates the symptoms of NASH and incidence of liver cancer [87]. E06 binds the PC headgroup of OxPLs but not to unoxidized phospholipids. The binding of E06 with oxidized phospholipids blocks the uptake of oxidized LDL by macrophages which inhibits the proinflammatory properties of OxPLs. Comparison of OxPLs content in the livers of AMLN-fed E06-scFv Ldlr−/− and Ldlr−/− mice shows that AMLN-fed E06-tscFv Ldlr−/− has reduced OxPLs [87]. In contrast, they exhibit similar levels of serum cholesterol and TGs. Furthermore, E06 significantly improves hepatic inflammation, steatosis, hepatocellular injury, and fibrosis. The progression from NASH to HCC was also reduced by neutralizing OxPLs in AMLN-fed E06-scFv Ldlr−/− compared with Ldlr−/− mice, as indicated by significant reduction in tumor number, tumor volume and tumor incidence. More studies have demonstrated inhibition of NAFLD using lecinoxoid, an oxidized phospholipid small molecule [88], while others showed that a combination of HFD and oxidized low-density lipoprotein promotes NAFLD [89]. Oxidized phospholipids promote these alterations as reactive oxygen species (ROS) bind to phospholipids of the mitochondrial membrane [90]. An increase in oxidative stress leads to a reduction in cardiolipins due to oxidation. Cardiolipins are mitochondrial membrane phospholipids that hold their structure and function to stabilize respiratory chain complexes and carrier proteins [91]. Oxidized cardiolipins destabilize the inner membrane and allow ROS and cytochrome c to leak out of the mitochondria, causing cell death [91]. These studies demonstrate an important role for oxidized phospholipids in the development of NASH and some mitigation factors include using antioxidant processes and antibodies that target and neutralize the oxidized phospholipid head and could potentially translate clinically with a similar naturally occurring antibody in humans.

## 6. Role of Cholesterol in NAFLD/NASH

In mammals, cholesterol is an important structural component of cell membranes and play a key role in membrane permeability and signaling processes [92,93]. Cholesterol serves as the precursor for all steroids hormones and bile acids [94]. The synthesis of cholesterol occurs in the cytoplasm and endoplasmic reticulum (ER) from acetyl-CoA via the mevalonic acid (MVA) pathway [94]. 3-hydroxy-3-methylglutaryl coenzyme A reductase (HMGCR) is the rate-limiting enzyme in the cholesterol biosynthetic pathway. In order to preserve a low sterol level, cholesterol and its precursors move out of the ER and attached to the cell membrane and participate in cellular signal transduction [95]. In addition, cholesterol is also converted into cholesterol esters by acyl-CoA acyl-transferase (ACAT) to reduce free cholesterol accumulation in the intracellular membranes and plasma [95]. Whole-body and cellular cholesterol homeostasis is a tightly regulated process in order to prevent accumulation of excess cholesterol or deficiency [96]. The liver plays a key role in the regulation of distorted cholesterol levels in the plasma through a complex metabolic process involving sterol transporters, lipoprotein and nuclear receptors, and transcriptional gene expression [97].

A disturbance in the cholesterol pathways that perform synthesis, transport and conversion of cholesterol is believed to play a significant role in developing liver damage [98]. The non-inflammatory intracellular fat disposition can result from imbalance between lipid disposition and lipid removal from the liver, which can lead to cirrhosis, hepatocellular carcinoma, and hepatic fibrosis [99]. Furthermore, ER controls cellular levels of cholesterol, and a protein misfolding in the ER can lead to irregular cholesterol synthesis and clearance, leading to NAFLD.

### Free Cholesterol Contribution to NAFLD/NASH

Extensive dysregulation of hepatic cholesterol homeostasis has been documented in NAFLD, leading to increased hepatic cholesterol levels, which occurs at multiple levels, including the increased hydrolysis of cholesterol esters into free cholesterol (FC), increased hepatic cholesterol synthesis, increased levels of active SREBP2, increased uptake of cholesterol-rich lipoproteins, and decreased cholesterol excretion in bile [100,101,102,103,104]. Hepatic inflammation has been shown to occur when there is an accumulation of cholesterol in the lysosomes of Kupffer cells (KC). Hydroxycholesterol (27 HC) can help move cholesterol from the lysosomes into the cytoplasm. Bieghs V. et al. showed that giving bone marrow transplants from irradiated Cyp27a−/−mice or wild type mice to mice that lack the expression of low-density lipoprotein receptor (Ldlr−/− caused hyperlipidemia [105]. In addition, subcutaneous injections of 27HC were administered to the Ldlr−/− mice and placed on either a chow or an HFC diet for 3 weeks. It was found that the mice who were transplanted with Cyp27a−/− cells showed a higher accumulation of cholesterol in KC cells than the WT, and the Cyp27a −/−transplanted mice also showed increased liver inflammation and damage [105]. The treatments with 27HC were sufficient to lower lysosomal cholesterol and reduce inflammation. Nonalcoholic steatohepatitis (NASH) and Type 2 diabetes have been associated with insulin resistance and dysregulated cholesterol within the liver. Van Rooyen D.M. et al. utilized Alms1 (foz/foz) and WT NOD.B10 mice to determine the relationship between insulin resistance and hepatic free cholesterol (FC) [106]. They found that HFD-fed mice developed diabetes and eventually NASH with fibrosis between 12 and 24 weeks, which is associated with an increase in hepatic cholesterol ester. HFD-fed foz/foz mice exhibit enhanced hepatic FC compared with WT mice and this was associated with upregulation of LDLR [106]. These studies suggest that hyperinsulinemia affects cholesterol homeostasis resulting in hepatic lipotoxicity and NASH. Other studies have shown that NAFLD and NASH occur due to accumulation of FC resulting from an increase in microRNAs (miRs), which further dephosphorylates HMGCR and increases its levels [98]. In addition, the FC triggers the production of Kupffer Cells (KC) and Hematopoietic Stem Cells (HSC) which leads to inflammation and fibrogenesis causing liver damage. The liver damage can progress to NAFLD and NASH [99]. One mechanism for the accumulation of FC is the overexpression of HMGCR [107]. LDL-cholesterol (LDL-C), a derivative of LDL-receptor (LDLR), is responsible for the feedback inhibition of HMGCR through the inhibition of SREBP-2 [108]. Sterol regulatory element binding protein 2 (SREBP-2) is an ER bound transcription factor that regulates intracellular cholesterol. SREBP-2 activates the expression of HMGCR and LDLR [1,108]. An observable decrease in LDL-C during NAFLD can account for the unregulated maturation of SREBP-2 and the subsequent activation of HMGCR [108]. Another mechanism that FC accumulation has an effect on NAFLD is by inducing mitochondrial dysfunction. An accretion of cholesterol within the mitochondria attenuates ATP synthesis and increases hepatocyte sensitivity to oxidative stress [99]. The overload of cholesterol also induces the unfolded protein response (UPR) in the ER [109]. Due to the induction of UPR, the JNK1 pathway is activated and contributes to hepatocyte apoptosis [110]. The majority of patients with NAFLD possess hypercholesterolemia, a condition that is characterized by high cholesterol levels resulting from an increase in hepatic production of non-HDL-C. It has been observed that hypercholesterolemia is caused by mutations that inhibit the ABCG5/ABCG8 heterodimer, one of the main facilitators of biliary cholesterol excretion [111]. It is suggested that the ABCA1 protein experiences protein degradation during NAFLD [112]. ABCA1 is a membrane transporter that carries cholesterol into the blood. This protein degradation would explain the decrease in cholesterol efflux and the subsequent overexpression of hepatic cholesterol [112]. Collectively, new approaches that target the nuclear receptors and the cholesterol metabolism pathways might become useful to reduce hepatic free cholesterol and liver injury associated with NAFLD and NASH.

## 7. Conclusions

There is extensive evidence to support a crucial role of membrane lipids in the development and progression of NAFLD and NASH. We discussed how different membrane lipid species that serve as an integral part of the cell membrane as well as in signal transduction to contribute to the pathophysiology of metabolic diseases. It will be very important to investigate whether there are gender-specific differences in the levels of different lipid species, as it relates to the pathogenesis of fatty liver disease since the prevalence of NAFLD is higher in males. The development of new genetic and pharmacological mouse models has shed light on the role of lipid species in physiology and disease. However, future work including basic and translational studies should focus on mechanisms of how various lipid species are transported to the cell membrane and novel strategies for targeting membrane lipid species as possible treatment for fatty liver disease and other metabolic diseases.

## Figures and Tables

**Figure 1 membranes-12-00410-f001:**
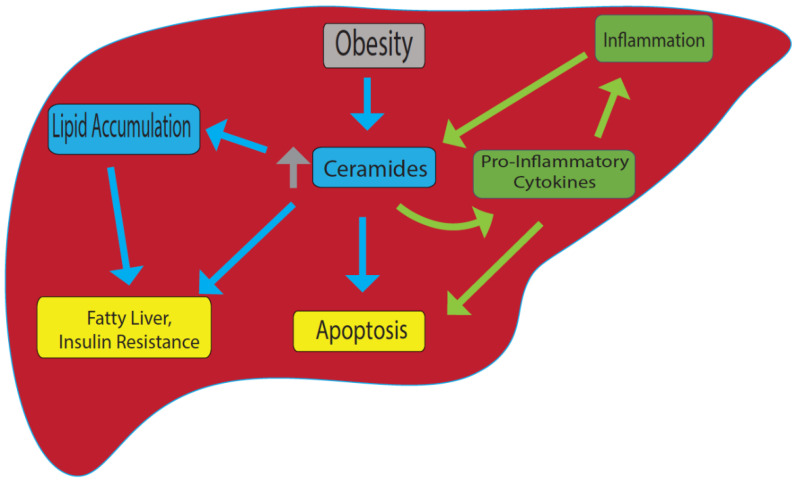
**Model for ceramide-induced fatty liver and other metabolic disorders**. In obesity, ceramide levels are upregulated, and this leads to accumulation of lipids, fatty liver and insulin resistance. On the other hand, inflammation can cause an increase in ceramide levels that promotes pro-inflammatory cytokine production. Increase in ceramide also cause apoptosis. Yellow: end results; Blue: linked specifically to obesity; green: linked specifically to inflammation.

**Figure 2 membranes-12-00410-f002:**
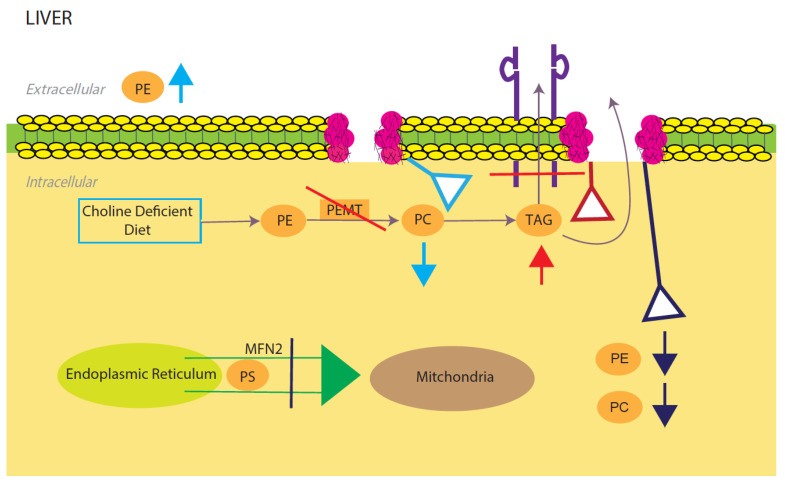
**Model for phosphatidylcholine (PC)/phosphatidylethanolamine (PE)-induced fatty liver disease.** Light blue, red, and dark blue are three different PE/PC membrane disruption pathways. Light blue shows the effects of a low choline diet, which significantly lowers intracellular PC levels due to lack of its primary reactant. The liver will try to compensate for this lowering by increasing extracellular PE. Red shows the effects of disrupting the PEMT pathway, which significantly decreases TAG secretion and increases TAG intracellular accumulation. Dark blue shows the effects of silencing the expression of mitochondrial protein Mfn2. In the ER, PS is converted to PE, which is then sent into the mitochondria for modification to PC. Knocking out this membrane protein causes a NAFLD/NASH phenotype due to the halting of PE and PC creation. All three disruptions upset the intracellular PE/PC ratio, which in turn causes membrane destabilization.

## Data Availability

Not applicable.

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
