# Peer review of "Characterization and Roles of Membrane Lipids in Fatty Liver Disease"

_membranes, 2022, doi:10.3390/membranes12040410_

Round 1

Reviewer 1 Report

This manuscript is well prepared, condensed and interesting form,  however drawings (figures) should be much more professional in form, made with the use of a graphic programme  (not drawing tools in editing programmes).

Author Response

April 5th, 2022

Mr. Henry Xu

Assistant Editor,

Membranes

Re: Characterization and Roles of Membrane Lipids in Fatty Liver Disease

Dear Henry,

I would like to thank you for the opportunity to address the reviewer’s comments on our recent submission to Membranes entitled “Characterization and Roles of Membrane Lipids in Fatty Liver Disease. As you will see from our responses to the reviewers’ all questions have been addressed. We think that the manuscript has been improved and we thank the reviewers for their comments. Below you will find a point-by-point response to all of the questions and comments raised by the reviewers.

Reviewer 1

Remarks on the revision I already noticed:

Q1. This manuscript is well prepared, condensed and interesting form,  however drawings (figures) should be much more professional in form, made with the use of a graphic programme  (not drawing tools in editing programmes)

  • We now use the graphic program Adobe Illustrator to draw the figures.

           In summary, we now hope that these figures will be satisfactory to the reviewers and that our manuscript be acceptable for publication in the Journal Membranes.

  Yours Sincerely,

 Ahmed Lawan, Ph.D.

Assistant Professor of Physiology

Department of Biological Sciences

Assistant Professor of Physiology

Department of Biological Sciences

Reviewer 2 Report

It is a nicely written manuscript that summarizes some of the findings concerning membrane lipids in nonalcoholic fatty liver disease (NAFLD). The collected data based on the currently published literature can expand our understanding of the characterization and their roles in pathogenesis of NAFLD with special regards to numerous membrane lipid species and their metabolites including phospholipids, sphingolipids and cholesterol. They discussed how dysregulation of their composition and physiology contributed to the development of fatty liver disease.

Novelty: Authors discussed a novel aspects of a crucial role of membrane lipids in the development and progression of NAFLD to nonalcoholic steatohepatitis (NASH).

Quality of presentation: Good.

Scientific soundness: The statement that it will be very important to investigate wheather there are gender specific differences in the levels of different lipid species as it relates to the pathogenesis of fatty liver disease since the prevalence of NAFLD is higher in males is still a key point of future scientific investigations. Future works including basic and translational studies should focus on mechanisms of how various membrane lipids are transported to the cell membrane.

Interest to the readers: This manuscript will be interesting for scientists (basic science) as well as clinicians.

Overall merit: Well prepared review.

Author Response

April 5th, 2022

Mr. Henry Xu

Assistant Editor,

Membranes

Re: Characterization and Roles of Membrane Lipids in Fatty Liver Disease

Dear Henry,

I would like to thank you for the opportunity to address the reviewer’s comments on our recent submission to Membranes entitled “Characterization and Roles of Membrane Lipids in Fatty Liver Disease. As you will see from our responses to the reviewers’ all questions have been addressed. We think that the manuscript has been improved and we thank the reviewers for their comments. Below you will find a point-by-point response to all of the questions and comments raised by the reviewers.

Reviewer 2

We thank the reviewer’s comments for his/her satisfaction with the initial submission of our manuscript. No remarks to be addressed.

           In summary, we now hope the manuscript will be satisfactory to the reviewers and that our manuscript be acceptable for publication in the Journal Membranes.

  Yours Sincerely,

 Ahmed Lawan, Ph.D.

Assistant Professor of Physiology

Department of Biological Sciences